# Custom-Made 3D-Printed Prosthesis after Resection of a Voluminous Giant Cell Tumour Recurrence in Pelvis

**DOI:** 10.3390/diagnostics13030485

**Published:** 2023-01-29

**Authors:** Adyb-Adrian KHAL, Dragos APOSTU, Calin SCHIAU, Nona BEJINARIU, Sebastien PESENTI, Jean-Luc JOUVE

**Affiliations:** 1Department of Orthopaedics and Traumatology, Iuliu Hatieganu University of Medicine and Pharmacy, 400000 Cluj-Napoca, Romania; 2Department of Paediatric Orthopaedics, AP-HM Timone Enfants, 13005 Marseille, France; 3Regina Maria Private Health Care Network, 400117 Cluj-Napoca, Romania; 4Department of Radiology, Iuliu Hatieganu University of Medicine and Pharmacy, 400000 Cluj-Napoca, Romania

**Keywords:** giant-cell tumour, pelvis, custom-made cutting guides, custom-made prosthesis

## Abstract

Giant-cell tumours are benign aggressive bone lesions that can affect any part of the skeleton. In early stages, curettage is preferred, but in case of local recurrence or voluminous lesions in the periacetabular region, wide resection and reconstruction are recommended. The purpose of this article is to increase clinicians’ awareness of the importance of the follow-up of these patients and to describe a case of a voluminous recurrence of a giant-cell tumour in the pelvis. We present a 25-year-old female who underwent internal hemipelvectomy assisted by 3D cutting-guides and reconstruction with a custom-made 3D-printed pelvic prosthesis, hip arthroplasty and ilio-sacral arthrodesis. No postoperative complications occurred and, at long-term follow-up, the patient had a stable and painless hip joint, good bone-implant osteointegration, with an excellent functional outcome. In spite of all available reconstructive techniques, in well-selected patients with voluminous pelvic resections, custom-made 3D-printed implants allow patients to have a good mechanical outcome.

## 1. Introduction

Giant-cell bone tumour is a benign lesion that generally occurs in young adults (65%), between the second and fourth decade of life with a clear female predilection [1,2]. It mostly affects long bones (65% in distal femur, proximal tibia, distal radius), but less frequently, they can occur in the sacrum or the pelvis [1,2]. Histopathologically, they are intramedullary lesions composed of mononuclear cells and osteoclast-like multinucleated giant cells, with a variable and unpredictable potential for growth and, sometimes, they can be very aggressive and destroy the cortex [1,3,4].

In early stages, surgical treatment mostly includes curettage, but the recurrence rate may range between 15–30% [1,4]. In the case of aggressive giant-cell tumours, wide resection is recommended but the major concern is the level of bone cut which must be accurately performed to achieve adequate margins (negative resection margins) and also to preserve as much bone stock as possible [5]. Extensive bone resection with safe margin is mandatory, otherwise local recurrence may occur [5]. Moreover, if follow-up in these patients is not regularly and properly conducted, sometimes, local relapse may be significantly more voluminous than the initial lesion [1,3].

Limb salvage surgery procedures on the pelvis requires appropriate excision of the lesion followed by careful reconstruction of the affected bones and soft tissues [6,7]. Pelvic resections and reconstructions are generally classified by tumour extension and the bone to be resected according to the Enneking classification [7]: type 1 involves the iliac bone, type 2 the periacetabular region, type 3 the pubis and type 4 the sacrum. In most lesions, the periacetabular localisation is often involved, which implies the replacement of the hip joint [8,9]. These surgeries remain challenging not only because of the complex and critical anatomical structures involved in the pelvis, but also in the hip.

Imaging-based new technologies found their way into the medical field and are of growing importance [10,11,12,13,14]. Three-dimensional-printing (3D-printing) techniques are now widely used in the resection and the reconstruction of bone tumours, including personalized bone cutting guides and customized implants [10,11,12,13,14]. Based on preoperative imaging, personalized bone cutting guides allow for safer and more precise resection margins, while personalized implants allow for restoration of patient-specific bone anatomy with a perfect matching between the implant and the host bone [10,11,12,13,14].

In this report, we describe the case of a patient with a voluminous recurrence of a giant-cell tumour after curettage in the pelvis, who underwent internal hemipelvectomy and reconstruction with a massive 3D-printed custom-made prosthesis.

## 2. Case Presentation

A 25-year-old female executive accountant was referred to our clinic with a right hip mechanical pain that was worsening during walking and daily activities. The patient presented, 6 years before, to a regional hospital for investigations after a traumatic incident. A plain radiography was performed and, incidentally, a well-delimited lesion was noticed (Figure 1). An open biopsy was done without any other investigations. The histopathological sample was suggestive for a giant-cell tumour. The local treatment was curettage and the bone defect was filled with bone cement. Two years later, local recurrence occurred and the patient visited several orthopaedic departments but treatment was refused due to the voluminous mass of the tumour and the lack of experience in the massive bone reconstruction field. The approximate dimensions of the bone tumour were about 10/7/6 cm (cranio-caudal/antero-posterior/latero-lateral), with an estimated tumour volume of about 210 cm^3^.

Our clinical examination revealed painful mobilities of the right hip, without evidence of a palpable mass. Hip mobilities were normal, except the external rotation which was limited to 25° and the abduction to 30° (active and passive). Normal walking was possible without any sign of gluteus medius insufficiency. The laboratory blood tests and urine analysis results were normal.

Radiography (Figure 2) revealed an expansive lytic bone mass centered on the right acetabular region, with cranial extension to the iliac wing, caudal to the ischium, and posterior nest to the lower third of the sacroiliac joint. A CT scan (Figure 3) demonstrated the expansive, lytic, lobulated bone mass, with few septa and small tissue areas, highlighting in more detail the marked osteolytic character, with several regions of interruption of the bone cortex, minimal periosteal reaction on the lateral aspect of the tumour. We also performed an MRI exam that revealed an intensely heterogeneous bone mass, with a small solid and relatively important cystic component (Figure 4) and small solid components with low to intermediate T1 signal, with moderate contrast enhancement. There was some enhancement of the cystic walls and septa. On T2-weighted images, there were heterogeneous high signals, with areas of low signal intensity due to hemosiderin or fibrosis. The T2 fat saturation axial images showed multiple cystic areas containing fluid–fluid levels, characteristics for an aneurysmal bone cyst component. In the surrounding bone marrow or soft tissue, no high signal suggestive of oedema was observed. MRI examination showed no tumoral lymph nodes in proximity.

We performed a percutaneous CT-guided biopsy and our diagnostic was positive for a giant-cell tumour recurrence.

Surgical treatment consisted of zone 1 and 2 Enneking internal hemipelvectomy (3D cutting-guides assisted) and reconstruction with a custom-made 3D-printed pelvic prosthesis, hip arthroplasty and ilio-sacral arthrodesis. To produce the 3D cutting-guides, we provided the company (Waldemar Link GmbH & Co. KG, Hamburg, Germany) with CT-scan (1.25 mm slices) and MRI of the patient. The surgeons and company specialists discussed possible layouts of the cutting-guides (for negative resection margins) (Figure 5a), implant and bone fixation until the final design was agreed (Figure 6a). Then, the company printed models of the affected bone, osteotomy-guides and implant (Figure 7b) in their natural dimension and size. These models were used to plan the surgery and to control if the cutting-guides were fitting correctly before using them on the patient. Manufacturing time of the agreed osteotomy-guides and implant was 6 weeks.

The patient was positioned in lateral decubitus and the approach was done through a large ilio-inguinal and ilio-femoral incision to expose both the internal and external aspects of the pelvis. The femoral nerve and the iliac vessels were reclined after dissection of the inguinal canal and disinsertion of the inguinal ligament. The internal sciatic notch, the internal gluteal vessels and ilio-sacral joint were exposed after extraperiosteal dissection of the ilium. The sartorius tendon, the rectus femoris tendon and the rectos femoris reflected tendon were detached. In the extrapelvic region, the fascia lata tensor muscle together with the gluteal muscles flap were detached from the iliac bone. A slight layer of muscles was kept in site in order to protect the planned negative resection margins. Dissection followed the gluteal arteries and the sciatic nerve and the external ilio-sacral joint was exposed. Anteriorly, the iliopectineal eminence and the obturator foramen were exposed. The obturator nerve and vessels were reclined and the sacrospinous ligament was released after hip joint capsulectomy and osteotomy of the femoral neck. Then, iliac and infraacetabular osteotomies and sacroiliac disarticulation were performed (Figure 5).

The resection sample was sent to radiological and histopathological exams (Figure 7).

The bone defect was reconstructed with a custom-made 3D-printed pelvic prosthesis that was fixed with 6.5 mm cancellous screws in the obturator quadratus and in the remnant iliac bone (Figure 6).

In addition, an 8.0 mm cancellous screw was added into the posterior part of the remnant iliac bone. Ilio-sacral fusion was performed using a 10 mm uncemented stem. In the end, the medius gluteus and the gluteus flap were reinserted on the iliac crest; the tensor fascia lata muscle, the sartorius muscle and the rectos femoris muscles were reinserted on the prosthesis. No intraoperative complications occurred.

The histopathological exam (Figure 8) revealed a proliferation of mononuclear cells with moderate atypia, scattered macrophages and large osteoclast-like giant cells. Mononuclear stromal cells are diagnostic and neoplastic components of tumour; no mitosis in this case. The proliferation had extensive secondary cystic changes—aneurysmal bone cyst-like, hemosiderin deposits. Tumour cells were negative for S100 and DOG1; giant cells positive for CD68. The resection margins were negative.

The operating time was 10 h and the estimated blood loss was 1200 mL. Intraoperatively, the patient received 1 red blood cell (RBC) unit, 1 RBC unit within the first 24 h postoperatively and 2 RBC units in the 2nd and 3rd day postoperatively. The patient stayed 3 days in the intensive care unit and 7 days in the conventional orthopaedic unit. A CT-scan was performed to confirm the implant position. The patient was discharged at postoperative day 10.

For the first 6 weeks, mobilisation was not allowed and a hip brace was applied. In the 7th week postoperatively, walking with crutches was permitted. Total weight bearing was allowed after 3 months postoperatively. Rehabilitation was conducted for 5 months in order to encourage full hip mobilities and to regain a normal gait while walking.

In the follow-up, the patient was evaluated every 6 weeks during the first 3 months and every 6 months until the first 2 years after surgery. Starting from the first postoperative consultation at 6 weeks, the hip joint was stable, the patient was painless and radiological examination showed no modifications. At the time of the last follow-up at 24 months, no modifications and no recurrence were observed (Figure 9). Functionally, the MSTS Score [15] was 28, with a minor limping. The patient returned to work 6 months after the surgery but this was related to the COVID-19 pandemic and not to the surgical intervention.

## 3. Discussion

Giant-cell tumours in pelvis are very rare (approximately 3%) [1,2]. Clinically, the pain dominates the scenario, but sometimes swelling or muscular hypotrophy may be encountered [1,2]. In our case, the lesion was discovered incidentally without any clinical reason, but latterly, when recurrence occurred, the symptoms were more evident.

There is still no widely accepted consensus on the ideal treatment for these lesions, but in early stages curettage is the preferred procedure, while in advanced stages, pathological fractures or in dispensable bones, wide resection is recommended [1]. Other treatment options include embolization, interferon, radiation therapy, aggressive curettage with phenol or cement filling or Denosumab [1,2,16,17,18]. However, some authors recommend neo-adjuvant treatments, such as Denosumab, prior to curettage in case of surgeries with difficult joint preservation [2,19,20]. It is believed that Denosumab treatment causes osteosclerosis of the lesion which may be surgically helpful in case of wide resection, but, in case of curettage, the lesion is more difficult to be resected leaving the tumour cells behind and, consequently, increasing the local recurrence rate [21,22,23].

The local recurrence rate after curettage rate may vary between 16–60% in case of Denosumab treatment, but even without, the rate ranges between 15–30% [19,21,22,23,24]. Sometimes, at the time of the relapse, the lesion may be more extensive and invasive, but this generally happens when the follow-up is not regularly conducted [1,19,21,22,23,24]. However, giant-cell bone tumours are locally aggressive by destroying the cortex; such voluminous lesions, as our patient had, represent the biological evolution in the absence of treatment or following curettage [1,2,3,19]. Our patient did not receive any local or systemic adjuvant or neo-adjuvant treatment and we did not know how aggressively the initial curettage was performed. However, we think that if a proper follow-up were performed, such voluminous local recurrence could have been avoided.

The surgical margins are of main interest both in benign aggressive and malignant bone tumours, because based on literature, patients with negative resection margins have better outcomes [5,13,24,25,26,27]. The recent development of custom-made 3D personalized specific instruments and implants provide solutions for this, especially in difficult tumour locations such as the pelvis [12,13]. When osteotomies are properly planned, the use of 3D cutting guides help to achieve proper tumour resection [13]. Moreover, 3D cutting guides allow to preserve as much bone stock as possible, in order to provide adequate structure of the future implant [12]. Identifying the correct resection border is the main key to achieve R0 resection (wide resection) and this way, the chance for local tumour recurrence is reduced [5,11,27]. In our study, at the 2-year follow up, we did not observe any local recurrence probably because of R0 resection using 3D cutting guides in this voluminous tumour.

Limb salvage for pelvic tumours represents a challenging surgical treatment for orthopaedic oncology surgeons because of the complex and critical anatomical structures involved in the pelvis and hip. Moreover, the major concern is that practicable reconstructive modalities have not been proved to be “ideal” because all of them are related to various complications regarding the functional outcome or mechanical failures such as deep infections, dislocations, implant loosening, graft/implant fracture, limb length discrepancy, etc. [8,9,12,16,26,27,28,29,30,31,32]. In some locations (iliac and iliosacral resections), the necessity of reconstruction of the pelvic ring is even controversial [29]. However, except in some oncological situations such as multi-metastatic sarcomas [9], reconstruction of the tumorous bone defects involving the acetabulum is recommended in order to restore acceptable limb function [10,12,14,30,32]. Many reconstruction techniques have been described following periacetabular resections: saddle prosthesis, allografts, allografts composite prosthesis, autografts and modular hip prosthesis or hip transpositions [10,12,14,30,32].

In recent years, thanks to computer-aided design and 3D-printed technology, reconstruction with perfectly matching implants in large bone defects after tumour resection became possible [10,11,12,13,14]. The purpose was to achieve an anatomical reconstruction by perfect matching between the implant and the host bone, better hip position (offset, rotation center and limb equality), better muscles reinsertion, lower complication rate, and, consequently, improved functional outcome [10,11,12,13,14]. In oncological surgery, many authors consider these implants to have a lower mechanical complication rate than any other biological reconstruction [10,33,34,35]. However, these implants are not exempt from postoperative complications. Deep infection or wound dehiscence are the most frequent issues [8,33,34,35]. However, these complications are related to invasive procedures with long surgical time [8,10,35]. For the correct positioning of the custom-made implant, the 3D cutting guides are fundamental. However, in case of delicate tumour locations, the surgery may be prolonged to perfectly adapt the 3D cutting guides according to the preoperative resection planning [10,11,12,13,14]. In our case, even if we had meticulously planned and calculated all the resection planes, the surgery was still long (10 h) mainly due to difficulties in fixing the distal (obturator) cutting guides and to perfectly match the customized implant in all three planes. Otherwise, we would have done intraoperative adjustments which were not possible because the osteotomy had to be performed according to the preoperative surgical plan.

Indications for custom-made implants after bone tumour resection in the pelvis are still debated [10,14]. Before choosing these implants as limb salvage surgery, Zoccali et al. [10] recommend considering several factors such as the patient’s age, the diagnosis, tumour location, etc. We also considered their decisional protocol [10] as a guide to proceed to limb salvage surgery with a custom-made implant in our patient. Being a young patient, complying with our future postoperative recommendations, without any timing stress regarding an oncological resection, and with no neo-adjuvant or adjuvant therapy such as chemotherapy or radiation therapy, allowed us to thoroughly plan the surgical procedure. Our patient is free of disease, without complications, without local recurrence and with an excellent functional outcome.

This study is a retrospective case report and, therefore, it was subject to inherent limitations and biases. The technique of reconstruction was not randomized, and the preference of the surgeon may have contributed to a selection bias. Second, it may not be possible to judge the true incidence of complications due to the limited sample size. In addition, many potentially uncontrolled variables existed, such as the amount of soft tissue excision and the characteristics of fixation. However, giant-cell tumours in the pelvis are rare lesions and we report a single case describing an entire procedure, with all steps from the medical imaging to the accurate resection and reconstruction.

## 4. Conclusions

Despite the low prevalence of giant-cell tumours in the pelvis, physicians need to be aware about this lesion that can be very aggressive and a thorough follow-up is mandatory in the case of curettage. In case of limb salvage surgery for pelvic tumours, personalized 3D-printed cutting guides provide solutions to remove the tumour with safe resection margins. It is our opinion that in well-selected patients with voluminous pelvic resections, custom-made 3D-printed implants are a good option to restore the bone and joint architecture and allow patients to have a good mechanical outcome and follow-up. Presently, these implants are common enough these days to be made available in oncological orthopaedic departments where these patients should be managed.

## Figures and Tables

**Figure 1 diagnostics-13-00485-f001:**
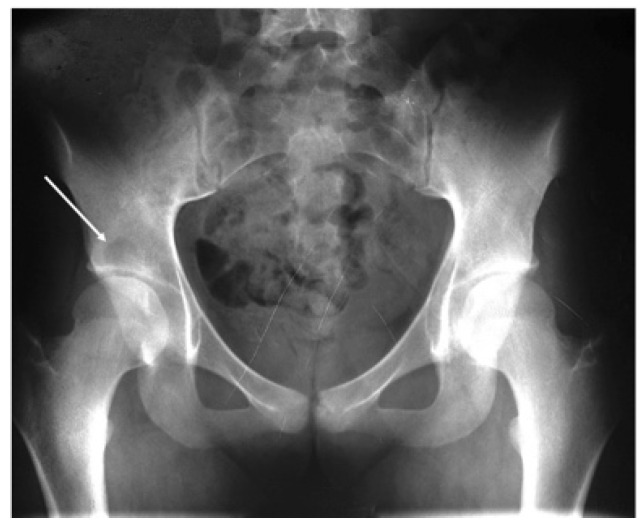
Pelvic AP radiography showing a well-delimited lesion (arrow) in the right supraacetabular area. The lesion is osteolytic without periosteal reaction or soft tissue invasion.

**Figure 2 diagnostics-13-00485-f002:**
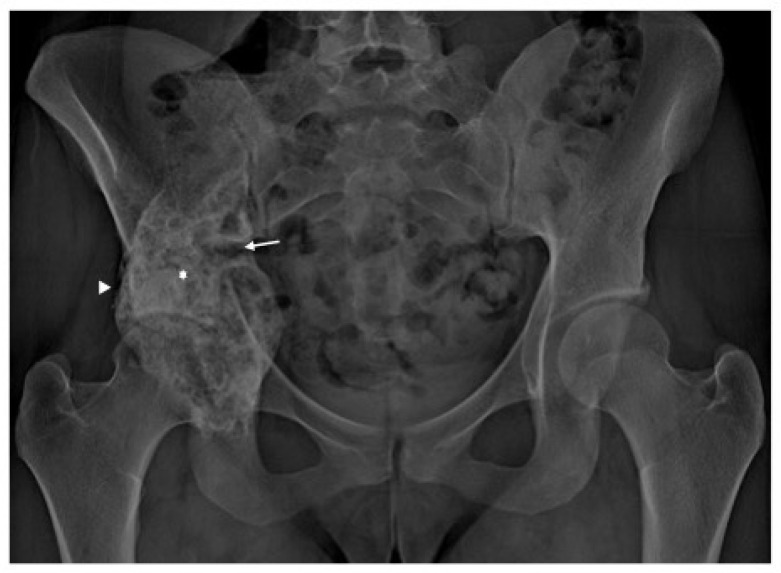
Plain radiography, frontal projection. Lobulated, mostly osteolytic, expansive, bone mass centered on the right acetabular region (star). Small areas of bone cortex interruption in the medial part (arrow). Minimal periosteal reaction in the lateral aspect (arrowhead).

**Figure 3 diagnostics-13-00485-f003:**
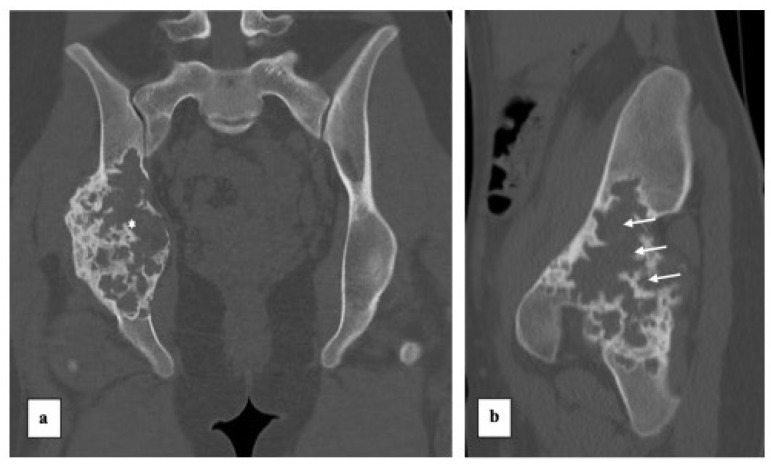
CT scan, coronal (**a**) and sagittal (**b**) views. CT scan reveals an intraosseous mass (star) centered on zone 2 Enneking, but with extension including zones 1 and 3. Marked cortical osteolysis, especially in the posterior region towards the sacroiliac joint (arrows).

**Figure 4 diagnostics-13-00485-f004:**
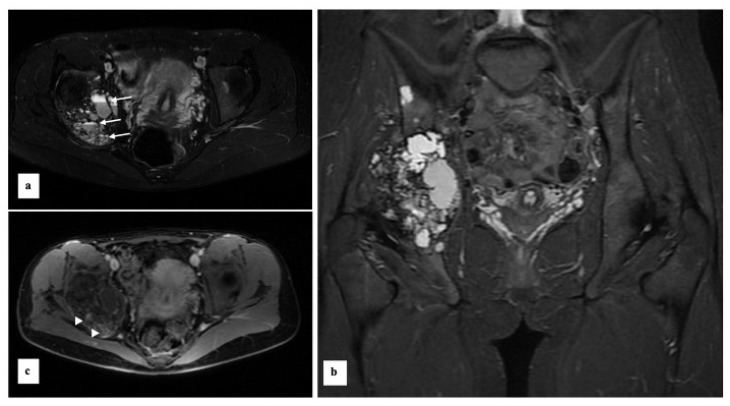
MRI axial (**a**,**c**) and coronal (**b**) views. T2 FS (**a**), STIR (**b**) and T1+k (**c**) sequences. MRI reveals an intraosseous heterogeneous mass. No adjacent bone marrow or soft tissue oedema. Relatively important cystic component with fluid-fluid level (arrows). Small solid components with moderate contrast enhancement (arrowheads).

**Figure 5 diagnostics-13-00485-f005:**
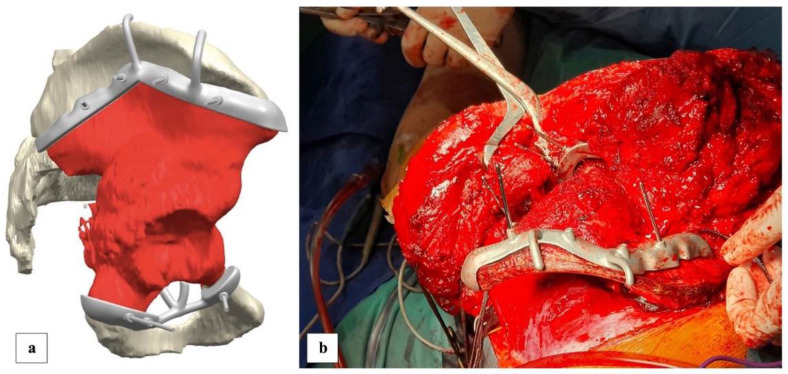
Preoperatory 3D setup of the cutting guides (**a**) were used to orientate the osteotomy of the bone tumour. Intraoperatory (**b**), the cutting guides were easily reproductible and this permitted us to obtain a negative margins resection of the bone lesion.

**Figure 6 diagnostics-13-00485-f006:**
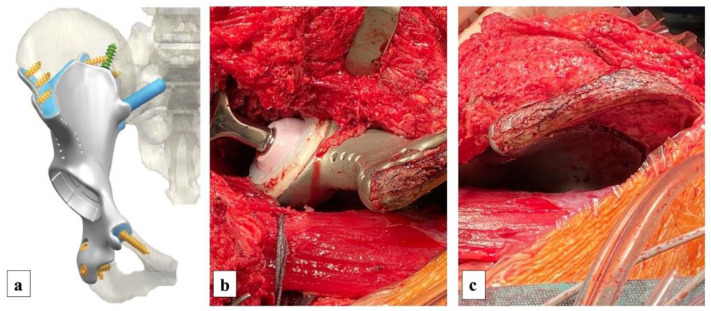
Preoperatory 3D setup of the custom-made implant (**a**) was used to analyse the possibilities of fixation to the host bone. In addition to the provided design, we asked for reinsertion holes for the tensor fascia lata muscle, the sartorius muscle, the rectos femoris muscle and the rectos femoris reflected tendon. After the reconstruction, the application of the implant was perfectly sealed to the host bone (**b**,**c**).

**Figure 7 diagnostics-13-00485-f007:**
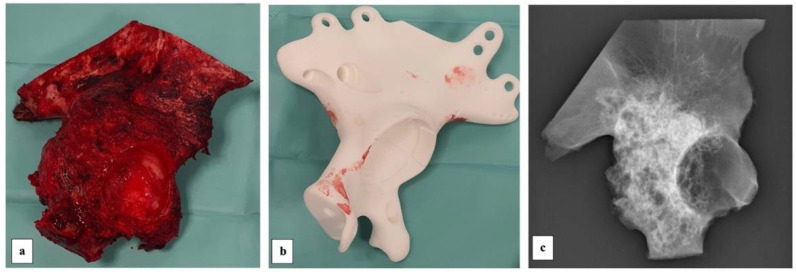
Image of the resection sample (**a**) that fitted in length and width with the preoperatory layout of the implant. This (**b**) may be used to assure the surgeon during the positioning of the cutting guides in case of difficulties. The resection sample was also radiographed (lateral view) (**c**).

**Figure 8 diagnostics-13-00485-f008:**
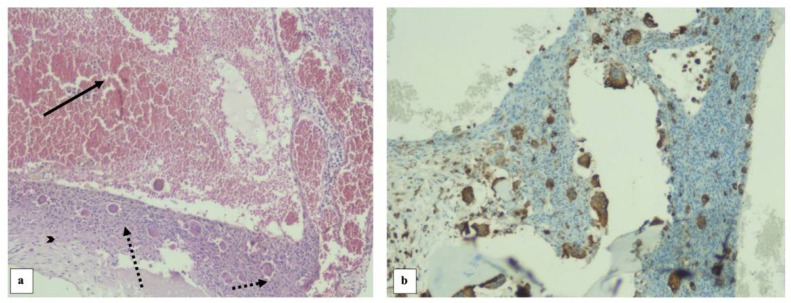
Histological specimen (**a**) showing large aneurysmal bone cyst-like changes (arrow), mononuclear cells (arrowhead) and osteoclast-like giant cells (round dotted arrows) in hematoxylin-eosin coloration at 100× magnification. Immunohistochemistry specimen at 100× magnification (**b**) was positive for CD68.

**Figure 9 diagnostics-13-00485-f009:**
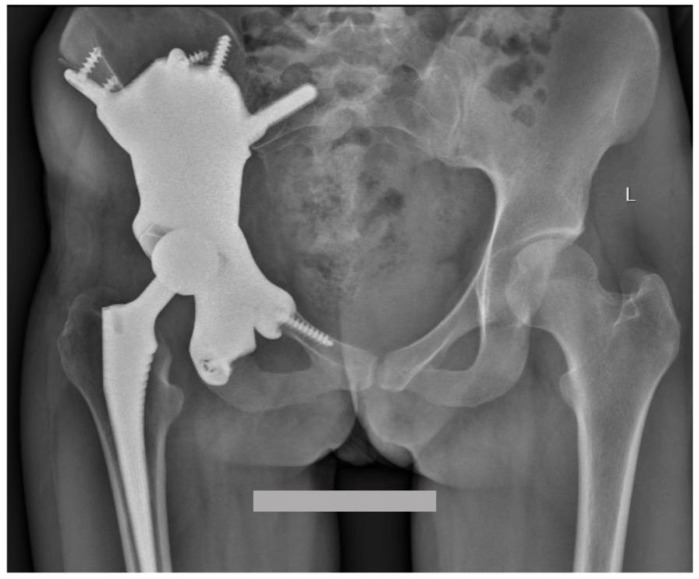
Pelvis AP radiography showing a custom-made 3D pelvic prosthesis in a 27 year old patient. At the last follow-up at 2 years, the implant was still in site with good fixation and no local recurrence of the giant-cell tumour was observed.

## Data Availability

On request from the corresponding author, the data are not publicly available due to privacy and ethical reasons.

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
