# Peer review of "Custom-Made 3D-Printed Prosthesis after Resection of a Voluminous Giant Cell Tumour Recurrence in Pelvis"

_diagnostics, 2023, doi:10.3390/diagnostics13030485_

Round 1

Reviewer 1 Report

Dear Author,

Is an interesting article that brings useful information for clinical practice.

I have a few observations.

Case presentation

Lines 63-64: You mentioned that „The symptoms started 6 years before and the patient presented to a regional hospital for investigations after a traumatic incident.”

And at Discussions

You mentioned that the symptoms appeared only after tumor recurrence.

Lines 189-190 “In our case, the lesion was discovered incidentally without any clinical accuse, but latterly, when recurrence occurred, the symptoms were more evident.”

Please clarify.

Kind regards,

Author Response

Article Title: Custom–made 3D–Printed prosthesis after resection of a voluminous Giant Cell Tumour recurrence in pelvis

Manuscript ID: diagnostics-2153981

Reviewer #1:

Reviewer(s)' Comments to Author:

Dear Author,

Is an interesting article that brings useful information for clinical practice.

I have a few observations.

Case presentation

  1. Lines 63-64: You mentioned that „The symptoms started 6 years before and the patient presented to a regional hospital for investigations after a traumatic incident.”

And at Discussions, you mentioned that the symptoms appeared only after tumor recurrence; Lines 189-190 “In our case, the lesion was discovered incidentally without any clinical accuse, but latterly, when recurrence occurred, the symptoms were more evident.”

Please clarify.

Thank you for giving us the opportunity to submit the revised draft of the Manuscript diagnostics-2153981, entitled „Custom–made 3D–Printed prosthesis after resection of a voluminous Giant Cell Tumour recurrence in pelvis” for publication in the journal of Diagnostics. We appreciate the time and effort that you dedicated to providing feedback on our manuscript and are grateful for the insightful comments on and valuable improvements to our paper. We have incorporated the suggestions made by the reviewer. Those changes are marked up using the “Track Changes” function (blue color). Please see below, for a point-by point response to the comments and concerns.

  1. Thank you for this precious observation. We modified the phrase accordingly.

Lines 72-74: Patient presented, 6 years before, to a regional hospital for investigations after a traumatic incident.”

Reviewer 2 Report

It is a good case report, providing valuable information and knowledge about giant-cell tumors in bones.

The main weakness of this manuscript is that, only a little information about the custom–made 3D–printed prosthesis used in this study is provided. For readers, the main value of this manuscript should be the information about the custom–made 3D–printed prosthesis, but this manuscript lacks such details. Hence, the reviewer suggests the authors must add more details about the custom–made 3D–printed prosthesis to the manuscript, regarding its manufacturing process, its technical parameters, and so on.

Regarding the writing: Throughout the manuscript, many sentences are too long and difficult to read, and many sentences have grammatical error. There are some typos as well. The reviewer suggests the authors must edit the manuscript for improving the English writing style and readability.

Please see below for some specific suggestions.

* Both the Abstract and Conclusions lack the descriptions of the main methodology, outcomes and conclusions. Please improve.

* Line 72: Please clarify the meaning of the term “10/7/6”.

* Line 120 and Line 146: The word “his” should be “hip”. Is it correct?

* Line 128: Please provide the company name.

Author Response

Article Title: Custom–made 3D–Printed prosthesis after resection of a voluminous Giant Cell Tumour recurrence in pelvis

Manuscript ID: diagnostics-2153981

Reviewer #1:

Reviewer(s)' Comments to Author:

It is a good case report, providing valuable information and knowledge about giant-cell tumors in bones.

  1. The main weakness of this manuscript is that, only a little information about the custom–made 3D–printed prosthesis used in this study is provided. For readers, the main value of this manuscript should be the information about the custom–made 3D–printed prosthesis, but this manuscript lacks such details. Hence, the reviewer suggests the authors must add more details about the custom–made 3D–printed prosthesis to the manuscript, regarding its manufacturing process, its technical parameters, and so on.

  1. Regarding the writing: Throughout the manuscript, many sentences are too long and difficult to read, and many sentences have grammatical error. There are some typos as well. The reviewer suggests the authors must edit the manuscript for improving the English writing style and readability.

  1. Please see below for some specific suggestions. * Both the Abstract and Conclusions lack the descriptions of the main methodology, outcomes and conclusions. Please improve.

  1. * Line 72: Please clarify the meaning of the term “10/7/6”.

  1. * Line 120 and Line 146: The word “his” should be “hip”. Is it correct?

  1. * Line 128: Please provide the company name.

Thank you for giving us the opportunity to submit the revised draft of the Manuscript diagnostics-2153981, entitled „Custom–made 3D–Printed prosthesis after resection of a voluminous Giant Cell Tumour recurrence in pelvis” for publication in the journal of Diagnostics. We appreciate the time and effort that you dedicated to providing feedback on our manuscript and are grateful for the insightful comments on and valuable improvements to our paper. We have incorporated the suggestions made by the reviewer. Those changes are marked up using the “Track Changes” function (blue color). Please see below, for a point-by point response to the comments and concerns.

  1. Thank you for this precious comment. According to your suggestions, we added information about the demand procedure of the implant, manufacturing process and delay of fabrication. We believe that including more technical information of the implant would not improve consistently the quality of the paper as the aim of this case report was to insist on the importance of the follow-up in patients who underwent curettage in order to avoid such voluminous recurrence. Also, we wanted to highlight the good mechanical outcome after reconstruction with massive 3D-printed custom-made prosthesis in very well-selected cases. Please, let us know if you think that more technical information (materials, consistency of the implant, technical specifications) except the newly-added one will be valuable for the manuscript.

The newly added text will be found at Lines 131-141: Surgical treatment consisted in zone 1 and 2 Enneking internal hemipelvectomy (3D cutting-guides assisted) and reconstruction with a custom-made 3D-printed pelvic prosthesis, hip arthroplasty and ilio-sacral arthrodesis. To produce the 3D cutting-guides, we provided the company (Waldemar Link GmbH & Co. KG, Hamburg, Germany) with CT-Scan (1.25 mm slices) and MRI of the patient. The surgeons and company specialists discussed possible layouts of the cutting-guides (for negative resection margins) (Fig 5a), implant and bone fixation until the final design was agreed (Fig 7a). Then, the company printed models of the affected bone, osteotomy-guides and implant (Fig 6b) in their natural dimension and size. These models were used to plan the surgery or to control if the cutting-guides were fitting correctly before using them on the patient. Manufacturing time of the agreed osteotomy-guides and implant was 6 weeks.”

  1. Thank you for your appreciation and comments. We tried to improve the manuscript by asking an English native speaker to check for grammatical and typographical errors in order to increase the quality of writing and readability. We checked the manuscript for repetition of phrases and we removed it from the text. All these modifications will be found throughout the text.

  1. Thank you for your appreciation and comments. We tried to improve the abstract and the conclusion in order to highlight the methodology of our case report and the good mechanical outcome of custom-made prosthesis. You will find in the manuscript the abstract, the conclusion and the limitations paragraph updated according to your suggestions.

  1. Thank you very much for your observation. We added the measurements parameters in order to clarify the dimensions of the lesion recurrence.

Lines 80 – 82: The approximate dimensions of bone tumour were about 10/7/6 cm (cranio-caudal/antero-posterior/latero-lateral), with an estimated tumour volume of about 210 cm3.”

  1. Thank you very much for your observation. We modified the phrase in order to avoid confusion regarding the rectos femoris reflected tendon.

Lines 147 – 148: The sartorius tendon, the rectus femoris tendon and the rectos femoris reflected tendon were detached.”

Lines 183 – 186: In addition to the provided design, we asked for reinsertion holes for the tensor fascia lata muscle, the sartorius muscle, the rectos femoris muscle and the rectos femoris reflected tendon.”

  1. Thank you very much for your comment. The company name was added to the manuscript.

Lines 133 – 135: “To produce the 3D cutting-guides, we provided the company (Waldemar Link GmbH & Co. KG, Hamburg, Germany) with CT-Scan (1.25 mm slices) and MRI of the patient.”

Round 2

Reviewer 2 Report

Most of the reviewer's comments had been addressed satisfactorily by the authors for improving the manuscript.

My last concern is about the abstract, which is neither standard nor satisfactory. A standard abstract should contain the descriptions of the Background, Materials and Methods, Results and Conclusion. However, the abstract of this manuscript only describes background and purpose. Please improve.

Author Response

Article Title: Custom–made 3D–Printed prosthesis after resection of a voluminous Giant Cell Tumour recurrence in pelvis

Manuscript ID: diagnostics-2153981

Reviewer #2:

Reviewer(s)' Comments to Author:

Most of the reviewer's comments had been addressed satisfactorily by the authors for improving the manuscript.

  1. My last concern is about the abstract, which is neither standard nor satisfactory. A standard abstract should contain the descriptions of the Background, Materials and Methods, Results and Conclusion. However, the abstract of this manuscript only describes background and purpose. Please improve.

Thank you for giving us the opportunity to submit the revised draft of the Manuscript diagnostics-2153981, entitled „Custom–made 3D–Printed prosthesis after resection of a voluminous Giant Cell Tumour recurrence in pelvis” for publication in the journal of Diagnostics. We appreciate the time and effort that you dedicated to providing feedback on our manuscript and are grateful for the insightful comments on and valuable improvements to our paper. We have incorporated the suggestions made by the reviewer. Those changes are marked up using the “Track Changes” function (blue color). Please see below, for a point-by point response to the comments and concerns.

  1. Thank you for your appreciation and comments. We tried to improve the abstract according to your suggestions:

Abstract: Giant-cell tumours are benign aggressive bone lesions that can affect any part of the skeleton. In early stages, curettage is preferred, but in case of local recurrence or voluminous lesions in periacetabular region, wide resection and reconstruction are recommended. The purpose of this case report is to increase clinicians’ awareness of the importance of the follow-up of these patients and to describe a case of a voluminous recurrence of a giant-cell tumour in the pelvis. We present a 25-year-old female who underwent internal hemipelvectomy assisted by 3D cutting-guides and reconstruction with a custom-made 3D-printed pelvic prosthesis, hip arthroplasty and ilio-sacral arthrodesis. No postoperative complications occurred and, at long-term follow-up, patient had a stable and painless hip joint, good bone-implant osteointegration, with an excellent functional outcome. In spite of all available reconstructive techniques, in well-selected patients with voluminous pelvic resections, custom–made 3D–Printed implants allow patients to have a good mechanical outcome.